# Metformin Influence on the Intestinal Microbiota and Organism of Rats with Metabolic Syndrome

**DOI:** 10.3390/ijms23126837

**Published:** 2022-06-20

**Authors:** Elena Ermolenko, Anna Simanenkova, Lyubov Voropaeva, Nadezhda Lavrenova, Maryna Kotyleva, Sarkis Minasian, Alena Chernikova, Natalya Timkina, Nikita Gladyshev, Alexander Dmitriev, Alexander Suvorov, Michael Galagudza, Tatiana Karonova

**Affiliations:** 1Federal State Budgetary Institution “Institute of Experimental Medicine”, 197376 Saint Petersburg, Russia; lu_bashka@mail.ru (L.V.); nadezhda.lavrenova.vrn@gmail.com (N.L.); mariha.lenivaya@mail.ru (M.K.); krinege@mail.ru (N.G.); admitriev10@yandex.ru (A.D.); or navivus2@gmail.com (A.S.); 2Almazov National Medical Research Centre, 197341 Saint Petersburg, Russia; annasimanenkova@mail.ru (A.S.); carkis@yandex.ru (S.M.); arabicaa@gmail.com (A.C.); n.timkina2014@yandex.ru (N.T.); galagudza@almazovcentre.ru (M.G.); karonova@mail.ru (T.K.)

**Keywords:** metformin, metabolic syndrome, impaired glucose tolerance, intestinal microbiome, myocardial infarction, cardioprotection

## Abstract

Metformin is a first-line drug for DM2 treatment and prevention, but its complex effect on impaired glucose tolerance (IGT), including its influence on myocardial resistance to ischemia-reperfusion injury, is not completely studied. We aimed to evaluate the influence of metformin on the intestinal microbiota (IM), metabolism, and functional and morphological characteristics of myocardium in rats with IGT. IGT was modelled in SPF Wistar rats with a high-fat diet and streptozotocin and nicotinamide injection. Rats were divided into three groups: IGT (without treatment), IGT MET (metformin therapy), and CRL (without IGT induction and treatment). IGT group was characterized by: higher body weight, increased serum glucose and total cholesterol levels, atherogenic coefficient, impairment in the functional parameters of the isolated heart during perfusion, and larger myocardium infarction (MI) size in comparison with the CRL group. IM of IGT rats differed from that of CRL: an increase of *Bacteroides*, *Acinetobacter*, *Akkermansia*, *Roseburia,* and a decrease of *Lactobacillus* genera representation. Metformin therapy led to the diminishing of metabolic syndrome (MS) symptoms, which correlated with IM restoration, especially with the growth of *Akkermansia* spp. and decline of *Roseburia* populations and their influence on other members of IM. The obtained results allow us to consider from a new point of view the expediency of probiotic *A. muciniphila* use for MS treatment.

## 1. Introduction

Metabolic syndrome (MS) is currently one of the most outstanding (important, or actual) problems of medicine. Importantly, MS with disorders of carbohydrate and fat metabolism can be considered as a set of modifiable risk factors for cardiovascular diseases (CVD) development [1]. MS components, such as obesity, arterial hypertension, dyslipidemia, and glucose metabolism impairment, including impaired glucose tolerance (IGT) and type 2 diabetes mellitus (DM2), not only often take place together but pathogenetically influence each other comprising so-called cardiovascular continuum. One of the drugs for the treatment of DM2 is metformin. It has been shown that this drug reduces gluconeogenesis, glucose absorption, secretion of glucagon-like peptide 1 (GLP-1), increases glucose utilization [2], activates the energy sensor 5′-adenosine monophosphate-activated protein kinase (AMPK) [3,4]. As previously described, the cardioprotective effect of metformin and its anti-diabetic action are connected with: (1) inhibition of complex 1 of the mitochondrial respiratory chain [5]; (2) AMPK activation [6]; (3) abolishing of oxidative stress-induced interactions between peroxisome proliferator-activated receptor alpha and cyclophilin D [7]. Metformin provides an effect by maintaining the integrity of the intestinal barrier by stimulating the production of short-chain fatty acids, in particular butyrate, regulating the metabolism of bile acids [2,8].

On the other hand, the effects of metformin may be connected with an indirect and direct effect on the gut microbiota. It has been proven that metformin can directly stimulate growth of *Bifidobacterium adolescentis*, *Subdoligranulum* sp. and *Akkermansia muciniphila* [9]. Moreover, metformin-dependent increases of *Escherichia* spp. and decreases of *Intestinibacter* spp. in the intestinal microbiome of people from different countries were observed [10]. At the same time, the analysis of the effectiveness of this drug on the microbiome in MS is difficult due to the lack of complete information about the features of the microbiota in its various manifestations.

The microbiota in DM2 is the most well studied (described in scientific literature). However, even in this case, contradictory data were obtained. For example, an increase in the ratio of *Bacteroidetes*/*Firmicutes* [11] and a decrease in this proportion was noted [11,12]. There was both an increase in representatives of the phylum *Bacteroidetes*, in particular its representatives: *Bacteroides* spp. [13,14,15,16,17,18,19], *Prevotella* spp. [16,18,20] and a decline of certain species of this genus: *B. coprophilus*, *B. dorei*, *B. vulgates*, *B. oleiciplenus* [21]. In addition, there was a decrease in the representation of the phylum *Firmicutes* [17,21,22,23], in particular, *Lactobacillus* spp. [14,24,25], *Enterococcus* spp. [20,25], *Lactococcus* spp. [10,26], *Roseburia* spp. [17,27], *Faecalibacterium* spp. [16,17,19] in DM2. There was also a decrease in the representation of the phylum *Actinobacteria* [17], in particular, *Nocardioides* spp. [20], *Bifidobacterium* spp. [20], *B. adolescentis*, *B. longuminfantis* [28]. The population of mucin-degrading *Akkermansia* spp., most often is reduced [13,26,29], however, this does not always happen [19]. Moreover, in comparison to DM2, the influence of metformin on microbiota in IGT is almost unknown.

Our aim was to evaluate the influence of multifunctional antidiabetic drug metformin on the body weight and metabolic parameters, including, glycaemia, and lipid profile, on the intestinal microbiota, as well as functional and morphological characteristics of the myocardium under ischemia-reperfusion conditions in rats with IGT. We also performed a correlation analysis between different parameters that underwent changes after IGT induction and exposure to metformin to establish a causal relationship and search for new aspects of metformin action.

## 2. Results

### 2.1. Body Weight

The results of comparing the weight of rats from different groups are presented in Figure 1. Initially, body weight did not differ significantly between groups. During the first 4 weeks weight gain in rats receiving HFD was slightly higher compared to rats fed with standard chow, though the difference was not significant statistically. IGT induction led to a significant body weight gain, which was thereafter observed in the IGT group up to the end of the experiment. Metformin administration to IGT rats caused a diminishing of body weight gain compared to IGT rats without treatment. Body weight in the IGT MET group was similar to that in the CRL group.

Attention was drawn to the tendency to decrease the weight of the omentum in the IGT MT group of rats. According to this parameter, differences were found only between the control group and the IGT group (Appendix A).

### 2.2. Isolated Heart Function and Myocardial Infarct Size

Changes in LVP taking place during the global ischemia are shown in Figure 2A. LVP was higher in the IGT group than in the CRL group with the most outstanding differences observed at the beginning of the ischemic period—on the 10th and 15th minutes. Metformin administration did not influence LVP, which was as high in the IGT MET group as in the IGT group.

The baseline LVDP values were similar in all the groups. LVDP in both IGT and IGT MET groups was higher than in the CRL group during the whole reperfusion period. Interestingly, no significant differences were observed between IGT and IGT MET groups (Figure 2A,B).

In accordance with the changes observed for LVDP, LVEDP in the IGT and IGT MET groups was lower than in the CRL group, with no significant difference between the IGT and IGT MET groups (Figure 2C). There was a tendency for a slight increase in LVEDP in the IGT MET group during the 75th and 90th min of the reperfusion period, but it did not reach the value typical for control animals (Figure 2C). Surprisingly, CFR values, being similar in all groups at baseline, was significantly higher in both IGT and IGT MET groups in comparison with the CRL group throughout the reperfusion period, with no difference between IGT and IGT MET groups (Figure 2D).

Myocardial infarction size in the IGT group was significantly larger than in the CRL group. Metformin administration to animals with IGT led to a decrease in myocardial damage size in comparison with the IGT group, MI size in the IGT MET group did not differ from that in the CRL group (Figure 3).

### 2.3. Biochemical Parameters

We analyzed blood biochemical parameters in rats—BGL and lipid profile (TC, TG, HDL, and LDL).

BGL in the IGT and IGT MET groups was significantly higher than in the CRL group directly after nicotinamide + streptozotocin injection. Thereafter BGL in animals with IGT not receiving treatment remained significantly higher than that in control rats, although not exceeding the normal value. Metformin administration to IGT rats lead to a BGL decrease—there was no difference in BGL in the CRL and the IGT MET groups (Figure 4).

Despite observed differences in BGL between the IGT and the IGT MET groups it seems important to emphasize that BGL mostly remained in the reference interval, therefore we can suggest that revealed the cardioprotective effects of metformin cannot be completely explained by glucose profile normalization.

Differences in the level of TG, HDL, LDL, CRI-1 and CRI-2 were not detected. However, total cholesterol (TC) levels (Figure 5A) and atherogenic coefficient (AC) (Figure 5B) were lower in group IGT and were fully recovered after metformin therapy.

### 2.4. Microbiome Study

#### 2.4.1. qPCR

The study was carried out by comparing the following microorganisms (the quantitative content of intestinal microbiota representatives): the total number of bacteria, *Acinetobacter* spp., *Citrobacter* spp., *Escherichia coli*, and enteropathogenic *E. coli, Proteus* spp., *Lactobacillus* spp., *Bifidobacterium* spp., *Bacteroides thetaiotaomicron*, *Bacteroides fragilis* group, *Clostridioides difficile*, *Clostridium perfringens*, *Enterococcus* spp., *Faecalibacteriumprausnitzii*, *Fusobacteriumnucleatum*, *Parvimonasmicra*, *Roseburia inilinivorans,* and *Akkermansia muciniphila*. The statistically significant results and trends obtained when comparing the microbiota of rats from different groups by qPCR are presented in Figure 6. A lower content of lactobacilli and a higher amount of *Acinetobacter* spp. were found in the IGT group. In the IGT MET group, the content of *Akkermansia muciniphila* was greater than after IGT induction without further therapy. After the introduction of metformin (IGT MET group), the content of *Acinetobacter* spp. differed from the control values to a significantly lesser extent. It should also be noted, that in the IGT MET group, the content of lactobacilli significantly less prominently differed from the CRL group compared with the IGT group. Attention was drawn to the tendency of the number of *Roseburiainulinivorans* to decrease after metformin administration.

#### 2.4.2. Metagenome (16S rRNA) Study

The representation of *Firmicutes* was greater in the IGT group and was equal to normal values after metformin administration (Figure 7A). The relative abundance of *Actinobacteria* was greater in the IGT MET group than in other groups (Figure 7B). Results of the metagenome study are presented at the genera (Figure 8) level.

The representation of *Akkermansia* spp. was higher in both study groups (IGT, IGT MET) than in the CRL group. However, the use of metformin led to an even greater increase in these bacteria populations (Figure 8A). The relative abundance of *Bacteroides* spp. increased in both IGT and IGT MET groups, but in the IGT MET group was significantly less prominent than in the IGT group (Figure 8B). The representations of *Roseburia* spp. (Figure 8C) and *Lachnospiraceae* family (Appendix A) was higher in both study groups (IGT and IGT MET) compared to the CRL group, being lower in the IGT MET group than in the IGT group.

### 2.5. Correlation Analysis between Intestinal Microbiomecontentand Other Parameters

The summary effects of IGT induction and metformin administration on metabolic and functional parameters and intestinal microbiome are presented in Appendix A as well as in Appendix A (Appendix A Section). In the correlation analysis, we focused on those bacterial taxa, whose quantity and representation changed after metformin treatment, as well as those biological parameters that are MS components and changed after treatment with this drug. 

#### 2.5.1. Correlation Analysis between Bacterial Taxa

Correlation analysis of the microbiota components with *Akkermansia* spp. revealed a positive correlation between the abundance of this bacteria and *Lactobacillus* spp., *Faecalibacterium* sp. and *Bifidobacterium* spp., as well as *Actinobacterium* relative abundance (Figure 9A). At the same time, there was a negative correlation between the taxa *A. muciniphyla* and *Acinetobacter* spp. quantity and *Lachnosperaceae* family in fecal samples. Correlation analysis of the microbiota components with *Roseburia* spp. revealed a positive correlation between this bacteria and the quantity of *Faecalibacterium* sp. and a negative correlation with *Actynobacteria* filum, *Bifidobacteroim*, *Lactobacillus,* and *Bacteroides* genera relative abundance (Figure 9B).

At the same time, a negative correlation relationship between *Acinetobacter* spp. and *Lactobacillus* (*r* = −0.52), *Bifiduobacterium* spp. (*r* = −0.41) and *Bacteroides* spp. (*r* = −0.57) quantity was revealed (Appendix A).

#### 2.5.2. Correlation Analysis between Bacterial Taxa and Metabolic Symptoms

##### Body Weight

Of great interest is the strong positive correlation between the body weight and both content (Figure 10) and relative abundance of *Roseburia inulinivorans*. This interrelation can partly explain the weight loss that accompanied the metformin administration and the downward trend in the considered bacterial taxa. Metagenomic analysis *revealed* a positive correlation between body weight and the phylum *Bacteroidetes* and genus *Bacteroides*. As expected, we revealed a negative correlation between weight and relative abundance of *Lactobacillus* spp.

The correlation analysis revealed patterns for interrelationships between intestinal microbiome representatives and BGL only in the metagenome study. A significant positive correlation was found between BGL and *Roseburia* spp. representations (Figure 11).

The myocardial infarction size was directly related to the abundance of *Roseburia* spp.

At the same time a negative correlation was detected between myocardial infarction size and bacteria belonging to the genus *Akkermansia* (*r* = −0.7), *Streptococcus* (*r* = −0.4), and *Butiricimonas* (*r* = −0.4) (Figure 12).

## 3. Discussion

We, like many other researchers, turned to the problem of treating MS, which is manifested by disorders of carbohydrate and fat metabolism, obesity, and the inevitable occurrence of cardiovascular pathology. A feature of this study was the use of the IGT model, which made it possible to assess the risks of myocardial infarction and attempts to link changes in the body that occur against the background of the use of the multifunctional drug metformin with changes in the composition of the intestinal microbiota.

In this work, a comprehensive experimental study of IGT itself and the metformin effect on the organism and its intestinal microbiota in IGT conditions was carried out. Both our team [29] and other authors [6,30] used this model of HFD and nicotinamide + streptozotocin injection for IGT modeling. Importantly, the use of this model led to the development of lipid metabolism disorders and body weight gain thus making this model relevant for MS representation. Moreover, IGT modelling caused functional and structural changes in the myocardium during ischemia-reperfusion injury similar to those observed in type 2 diabetes mellitus, proving the hypothesis that even prediabetes provokes myocardial damage and increases cardiovascular risk [29]. Metabolism impairment started to develop on the 2nd–3rd day after nicotinamide + streptozotocin injection. It should be noted, that BGL in the IGT group was significantly higher than in the CRL group until the end of the experiment. Lipid metabolism disorders were confirmed in the IGT group by an increase in total cholesterol and AC at the end of the experiment. At the same time, functional and morphological disorders of the myocardium were revealed in ischemia-reperfusion injury of the isolated heart of IGT rats. Similar changes in the myocardium have already been noted earlier when using models of antibiotic-associated dysbiosis [31] and obesity [32,33]. The changes that the authors attributed mainly to changes in the microbiota (excessive growth of opportunistic enterobacteria), an increase of C reactive protein, and pro-inflammatory cytokines level in blood serum were revealed [32]. So it was shown, that the changes in myocardium could be associated not only with the MS complications but also with changes in the microbiota caused by the administration of antibacterial drugs: ampicillin, metronidazole, azithromycin, tetracycline, leading to intestinal dysbiosis [31,32,34]. A significant contribution of the microbiota to the development of CVD was proved by the effectiveness of probiotic bacteria (enterococci, saccharomycetes, and lactobacilli) used for dysbiosis correction [31]. Microbiota normalization in these experimental models correlated with a decrease in inflammation.

The features of the intestinal microbiota revealed by us in experimental IGT were in many respects similar to those previously described in animals and humans with DM2 by other studies. However, in many ways, they differed from them. It could be both due to the peculiarity of experiment design and the study of the initial stages of development of more severe pathology (DM2).

Unlike other reports, describing the microbiota in DM2an increase [13,18,23,35] or decrease [21,22] in the representation of the phylum *Bacteroidetes* were not detected. At the same time, an increase in the representation of the *Firmicutes* phylum was revealed, as noted earlier [22]. We can suspect that such parameter as *Bacteroidetes*/*Firmicutes* ratio is not reliable enough for assessing pathognomonic shifts in MS, in particular in IGT. At the same time, the increase in the representation of the genus *Bacteroides* attracted our attention. This contradicts data received earlier, evidenced by decrees of these bacteria populations in MS [21,22]. It should be noted, that the decline in *Bacteroides* spp. representation in the intestinal metagenome could be associated with insulin resistance and dyslipidaemia [36]. In our experiment, such a correlation was not established.

The greatest surprise was an increase in the content of butyrate-producing *Roseburia* spp. and *Akkermansia* spp., the representation of which is usually reduced in DM2 [13,17,26,27]. This phenomenon might be connected with compensatory changes in the microbiota in response to the development of more serious pathology than IGT.

Our attention was drawn (to the increase in the quantitative content of opportunistic *Acinetobacter* spp., which can be considered as a potential cause of so-called “low-grade inflammation” development that often accompanies MS [37]. The signs of low-grade inflammation detected when using this model earlier [31] were not evaluated in this experiment.

There was also a confirmed microbiota profile typical for DM2 [18,38] a decrease in the lactobacilli content. It was unexpected, because *Lactobacillus* spp. can normalize weight, lipid metabolism (decrease LDL/HDL and cholesterol),and glucose level [39].

We administrated metformin orally because not intravenous metformin administration has glucose-lowering properties [40]. This phenomenon suggests that the intestines might be one of the main target organs for metformin action [41]. Metformin realizes both its direct and pleotropic, including cardiovascular, effects by means of different mechanisms. In particular, it is known that metformin can modulate glucagon-like peptide-1 (GLP-1) receptor expression in the pancreas and increase plasma GLP-1 concentrations [42,43], possibly through enteroendocrine L-sells stimulation inside the intestines. The clinical and experimental trials revealed cardioprotective and anti-inflammatory properties of glucagon-like peptide-1 (GLP-1) analogs like liraglutide (Lira). demonstrated improved survival in lipopolysaccharide (LPS)-induced endotoxemia by inhibition of GLP-1 degradation [44].

Moreover, metformin can realize its actions in close connection with gut microbiota compound changes. In our study, the changes in intestinal microbiome coincided with the data obtained by other researchers [10,45] only with respect to the increase of *Akkermansia* spp. population and the decrease of members of the *Enterobacteriaceae* family.

We have shown in the IGT model for the first time a complete recovery of *Firmicutes* phylum and a sharp decrease in *Acinetobacter* sp. and a seemingly illogical representation of *Bacteroides* and *Roseburia* genera additionally.

The increase in the akkermansia population in the IGT model after metformin turned out to be important. It was unexpected for us. Positive metabolic and pleotropic effects of *Akkermansia muciniphila* were established earlier. *A. muciniphila* abundance is reduced in individuals with irritable bowel disease and other chronic diseases like obesity, as well as DM2 and prediabetes [38]. Thus, akkermansia growth is associated with body weight reduction, as well as hepatic steatosis, inflammation decrease, and even dyslipidemia reduction leading to atherosclerosis risk decrease. *Akkermansia* spp. also improves insulin sensitivity thus positively influencing type 2 diabetes mellitus pathogenesis [46]. *Akkermansia muciniphila* growth has been observed in obese people with healthier metabolic status and less prominent cardiovascular risks [47]. At the same time, the mechanisms of action of *Akkermancia* spp. are currently not entirely clear.

*Akkermansia* spp. are commensal Gram-negative bacteria, and are members of the phylum *Verrucomicrobia* [28]. *A. muciniphila* being Gram-negative bacteria, *A. muciniphila* also contains lipopolysaccharide endotoxin in its outer membrane of the cell wall, which differs from that of *Escherichia coli* [48]. Moreover, it was shown in a mouse model, that the presence of these bacteria in the intestine does not increase, but even reduces LPS-endotoxinemia in animals with a long-term fat diet [45]. These bacteria break down mucins and convert them into short-chain fatty acids (SCFAs), including acetate and propionate. Acetate is used by other beneficial bacteria, such as *Firmicutes*, to produce butyrate, a vital energy source for the cells that line the intestinum [41]. The latter gastrointestinal mucins that are able to protect the epithelium from pathogens [49].

In our t study a negative correlation between the quantitative content of *A. muciniphila* and *Acinetobacter* spp. was revealed, which suggested the presence of mechanisms for suppressing the reproduction of opportunistic bacteria in the organism by these bacteria.

The quantitative content of *A. muciniphila* correlated directly with the number of bifidobacteria, lactobacilli, which are usually considered as obligatory intestinal microbiota members and active against pathogens. This was confirmed by the presence of a negative correlation between *Lactobacillus* spp. and *Bifidobacretium* spp. with *Acinetobacter* spp. quantity.

Indirectly, the beneficial role of *A. muciniphila* against the background of metformin therapy IGT was supported by a negative correlation with glucose levels and an increase of resistance to myocardial ischemia. A negative correlation with the level of serum glucose was also established for the *Actinobacteria* phylum (bifidobacteria and other), the growth of which could be stimulated by *A. muciniphila*.

The direct and indirect effect of akkermansia on the level of glucose has been proven phenomenologically and can be explained by the stimulation of the growth of bacteria utilizing glucose, influence on GLP-1 expression and regulation of the absorption process at the level of the intestinal mucous membrane.

If we consider the patterns of changes in the composition of the microbiota and the symptoms of MS (weight loss, glucose levels, and resistance to cardiac ischemia), then we should consider another bacterial genus. Roseburia, belonging to the *Lachnospiraceae* family, which demonstrated a negative correlation with the representation of *Akkermansia* spp. Unlike akkermansia, the decrease in the quantity and representation of these bacterium correlated with an increase in the populations of *lactobacilli*, representatives of the phylum *Actinobacteria*, *Bifidobactrium* spp. and *Propionibacterium* spp. as well as a decrease in the population of *Fecalibacterium* sp. They also had a negative correlation with the weight of the animals, the level of glucose, and the infarction size. Consequently, roseburia, in its influence, in fact, could be an antagonist of akkermansia and in the case of an increase of its population, could counteract the akkermansia action.

*Roseburia* spp. are commensal bacteria, members of the phylum *Firmicutes*, *Lachnospiraceae* family. They produce short-chain fatty acids, especially butyrate, affecting colonic motility, immunity maintenance, and anti-inflammatory properties [36,50]. It was shown, that the increased abundance of *Roseburia* is associated with weight loss and reduced glucose intolerance in mice [51]. Apparently, other bacteria, in particular, *Akkermansia* spp., KM decrease in the intestinal microbiome.

Separately, we should dwell on the previously discovered [29] cardioprotective effect of metformin confirmed in our study by the decrease of myocardial infarction size in risolated rat hearts subjected to global ischemia followed by reperfusion. The mechanisms responsible for metformin cardioprotective effect realization may include influence on lipid profile, anti-atherosclerotic action, vascular endothelium protection, and influence on vascular smooth cells [52]. In our study, for the first time, causal relationships between the presence of the cardioprotective effect of metformin with an increase in individual representatives of the microbiota, primarily akkermansia were identified. However, the nature of this positive influence is not yet clear and only opens the way for further research.

Despite the fact that type 2 diabetes mellitus is accompanied by a severe increase in cardio-vascular risk, leading to increased incidence of myocardial infarction and even cardiovascular death. On the other hand, data concerning IGT’s influence on cardiovascular morbidity and mortality is even less presented in the scientific literature. Moreover, IGT’s impact on myocardial damage size is even less studied. Moreover, recently much attention has been paid to the hypothesis that pleotropic cardioprotective effects of metformin in diabetes mellitus could be connected with metformin influence on microbiota compounds. On the other hand, a similar effect in IGT conditions remains the subject of investigation, which, in our opinion, forms the novelty of our study. Our study was the first to investigate correlation links between myocardial damage and microbiota compound in IGT conditions which helps to elucidate potential mechanisms of metformin cardioprotective effect.

The strong side of our study is the correlation analysis between microbiota features and MS symptoms, which opens up new areas of research already more complete, taking into account the dynamics of MS symptoms changes, the impact on the digestive, cardiovascular systems, immune and may be other (nerves and endocrine) systems.

## 4. Material and Methods

### 4.1. Animals

The study was carried out in 38 male Specific pathogen-free (SPF) Wistar rats weighing 150–250 g (6–7 weeks old) obtained from the Animal Breeding Center, (Puschino, Russia).

The animals were maintained in fixed light mode, 12.00:12.00 h (light:dark), with no more than 5 animals per cage with free access to food and water. The temperature was maintained within the range of 22–25 °C, and the relative humidity—50–70%.

The duration of quarantine (acclimatization period) for all animals was 14 days. During the quarantine, every animal was examined daily. The color of the skin and visible mucous membranes, behavior, the motor activity, the presence of seizures, changes in respiratory movements, and tail position were assessed. The weighing was carried out upon arrival of the animals and during the quarantine period—at least once a week. Animals with deviations in weight, general condition, or behavior were not included in the experiment.

### 4.2. Ethics Approval

All experimental procedures were performed in accordance with the Guide for the Care and Use of Laboratory Animals (NIH publication No. 85–23, revised 1996) and the European Convention for the Protection of Vertebrate Animals used for Experimental and other Scientific Purposes. The study protocol was approved by the Institutional Animal Care and Use Committee of Almazov National Medical Research Centre (Protocol Number 19-1П3#V1, accessed on 14 January 2019). All efforts were performed to protect the laboratory animals and minimize their suffering throughout the study. The experiments complied with the ARRIVE guidelines (http://www.nc3rs.org/ARRIVE accessed on 14 January 2019).

### 4.3. Induction of IGT

Animals (except for the control group) were kept on a high-saturated-fat diet with (BioPro, Novosibirsk, Russian Federation: exchange energy 2690 kcal/kg, crude protein 20%, crude fat 22%) [30] (further: high-fat diet, HFD) throughout the experiment.

After the first 4 weeks of HFD, a solution of nicotinamide 230 mg/kg was injected intraperitoneally as a pancreatic protector, followed by an intraperitoneal injection of a solution of streptozotocin 60 mg/kg as a pancreatic toxin 15 min later [53].

### 4.4. IGT Verification

To verify the sufficiency of IGT modelling, blood glucose levels (BGL) in IGT and IGT MET groups were measured on the 2nd and 3rd day after nicotinamide and-streptozotocin injection. Nicotinamide (Sigma-Aldrich, St. Louis, MO, USA) 230 mg/kg and streptozocin (Sigma-Aldrich, St. Louis, MO, USA) 60 mg/kg were injected intraperitoneally.

Glycemic values of 3.3 to 7.8 mmol/L were considered normal—these rats were excluded from the experiment. If BGL reached 11.1 mmol/L DM2 during two consequent days was diagnosed [54] and these animals also were not included in the future experiment. If BGL on the second and third day was in the range of 7.8–11.0 mmol/L an oral glucose tolerance test was performed. BGL was measured initially (fasting) and 15, 30, and 60 min after gastric administration of a 40% glucose solution 3 g/kg. We diagnosed IGT if BGL was in the interval of 7.8–11.0 at least at one time point (15, 30, or 60 min after glucose solution administration) and was not equal and did not exceed 11.1 mmol/L ever [55].

### 4.5. Study Design

After the acclimatization period, the following experimental groups were formed. Control (CRL) group of rats (n = 20) fed with standard chow for 16 weeks. The impaired glucose tolerance (IGT) group of rats (n = 6) was kept without treatment for 12 weeks after the induction of IGT; IGT MET group of rats (n = 12) was treated with metformin (8-weeks) for 4 weeks after the induction of IGT. Metformin powder (Metformin hydrochloride Sigma-Aldrich, St. Louis, MO, USA) was dissolved in distilled water and given per os by gastric tube 200 mg/kg of body weight.

The animals from IGT and IGT MET groups were receiving HFD throughout the entire experiment. The study design is presented in Table 1.

Once every two days during the entire experiment, the animals were weighed, and the weight of the chow consumed in 2 days was determined.

Glucose measurement was performed using StatStrip Xpress Glucose/Ketone Meter (Nova Biomedical, Waltham, MA, USA). The glucose assessment in all groups was performed at the end of the 4th week, 8th week, and once a week during the weeks 9–16 at the same time of time (not after fasting).

Another biochemical study of blood serum samples was performed automatically by Abbot ARCHITECT si 8200 (333 Fiske Street, Holliston, MA, USA). The following parameters were studied: total cholesterol (TC), triglycerides (TG), high-density lipoproteins (HDL), and low density lipoproteins (LDL). The analysis of the lipid profile parameters ratio was carried out [56]. The atherogenic coefficient (AC) and two Castelli risk indexes (CRI)we calculated by following formulas: AC = (TC − HDL)/HDL; CRI-1 = TC/HDL and CRI-2 = LDl/YDL.

### 4.6. Isolated Heart Perfusion According to Langendorff

After reaching the surgical stage of anesthesia (tiletamine hydrochloride 30 mg/kg and zolazepam hydrochloride 30 mg/kg IM, then xylazine hydrochloride 6 mg/kg IM), thoracolaparotomy was performed to remove the heart. Then the heart was connected to a modified Langendorff apparatus (Appendix A). Retrograde perfusion was performed through the ascending aorta with modified Krebs-Henseleit buffer solution, consisting of the following (in mmol/L): NaCl 118.5; KCl 4.7; NaHCO_3_ 25.0; KH_2_PO_4_ 1.2; MgSO_4_ 1.2; glucose 11.0; CaCl_2_ 1.5) at a constant pressure of 80 mm Hg and at +37 °C [57]. After the end of the stabilization period (5 min), the following parameters were registered: left ventricular systolic pressure (LVSP), left ventricular end-diastolic pressure (LVEDP) that were evaluated iso volumetrically using a nonelastic polyethylene balloon introduced into the left ventricle, left ventricular developed pressure (LVDP) was calculated as the difference between LVSP and LVEDP. Coronary flow rate (CFR) was evaluated by measuring the time for the collection of perfusate outflow.

Global 30-min normothermic myocardial ischemia followed by 90-min reperfusion was induced by the reversible shutdown of perfusion. During the ischemic period, intra-left ventricular pressure (LVP) was recorded every 5 min in order to assess the severity of ischemic contracture. Ischemic contracture was determined as at least a threefold increase in LVP at any time during the ischemic period compared to the LVP after five minutes of ischemia. During the reperfusion period, functional parameters were recorded every 15 min, as done previously [29,58].

### 4.7. Myocardium Infarction Size Measurement

At the end of reperfusion, the volume of the irreversibly damaged myocardium was measured using the method of histochemical staining of transverse heart sections with 1% triphenyltetrazolium chloride solution. The sections were incubated in the indicated solution for 15 min and the viable myocardium was stained bright red, while areas of irreversibly damaged myocardium remained unstained. The ischemia size was recorded according to the method described earlier [29]. Infarction size was expressed as a percentage of total ventricular area minus the cavities.

### 4.8. Study of the Gut Microbiota

#### 4.8.1. Quantitative Polymerase Chain Reaction

A quantitative polymerase chain reaction (qPCR) was performed using the kit Colonoflor (AlphaLab, St. Petersburg, Russia) corresponding to the set of marker colonic bacteria on the qPCR unit Mini-Opticon, BioRad (Applied Biosystems, 850 Lincoln Centre Drive, Foster City, CA, USA) [59].

#### 4.8.2. Metagenome (16S rRNA) Study

Fecal samples frozen on the day of the material’s collection were used for the metagenome analysis. Libraries of hypervariable regions, V3 and V4, of the 16S rRNA gene were analyzed on MiSeq (Illumina, San Diego, CA, USA). DNA was isolated from feces using the kit Express-DNA-Bio (Alkor bio, St Petersburg, Russia). The standard method recommended by Illumina based on employing two rounds of PCR was used to prepare the libraries. The first round was the amplification of a 16S rRNA gene fragment, adding to the target fragment of the adapter nucleotide sequences included in the PCR primers [60].

Fastqc was used to assess the quality of raw reads. CD-HIT-OTU-Miseq was used for OTU retrieval with the following parameters: lengths of high-quality parts of R1 and R2 reads of 200 and 180 bp respectively, 97% read similarity for clustering cutoff, and 0.00001 for abundance cutoff. OTUs were annotated using Greengenes database version 13.5 [61]. CD-HIT-OTU-Miseq allows you to select OTU by the sequence of end sections without the need to merge paired sequences. R package phyloseq was used to assess class abundances and beta diversity.

### 4.9. Statistical Analysis

The normality of the data distribution was determined using the Kolmogorov–Smirnov criterion. The nonparametric criteria were used on this basis. Statistic data processing was performed using the software package IBM SPSS Statistics-22 (IBM, Armonk, NY, USA) and Statistica-10 (StatSoft Inc., Tulsa, OK, USA). The presence of statistically significant differences among groups was determined using the Mann–Whitney U-criterion, adjusted for multiple comparisons by the Benjamini–Hochberg method. Wilcoxon’s t-test was also used for paired samples. Comparative analysis was conducted using the a posteriori test honestly significant difference (HSD) for unequal N in the Statistica-8 software (StatSoft, Tulsa, OK, USA). Search for correlations between the studied parameters was performed using Spearman’s test using the software package Statistica 10.0 (StatSoft, Tulsa, OK, USA). Differences in *p* < 0.05 were considered significant.

## 5. Conclusions

The use of metformin in experimental IGT models led to several positive effects: weight loss, a decrease of glucose and total cholesterol levels in the blood, increased resistance to myocardial ischemia (decreased MI size), and was accompanied by the changes in the intestinal microbiota. The most important changes in the intestinal microbiome were the following: an increase in *Akkermansia* spp. and a decrease in *Roseburia* spp. quantity content and relative abundance. These findings allow suggesting, that the positive influence of metformin on the condition of patients with MS is related to the changes in microbiota, especially in *Akkermansia* and *Roseburia* genera populations. This should be taken into account when determining indications for the administration of probiotics based on these bacteria for the treatment of MS and prevention of its cardiovascular complications.

## Figures and Tables

**Figure 1 ijms-23-06837-f001:**
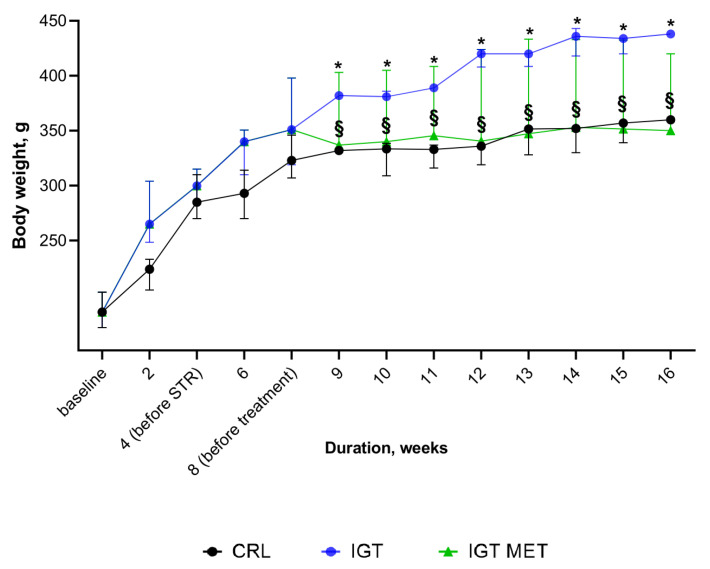
Dynamic of body weight in study groups. Notes: C- CRL (control group), IGT—impaired glucose tolerance group, IGT MET—impaired glucose tolerance + metformin group. Results are presented as median (25%; 75%). * *p* < 0.05, in comparison with CRL group, § *p* < 0.05, in comparison with IGT group.

**Figure 2 ijms-23-06837-f002:**
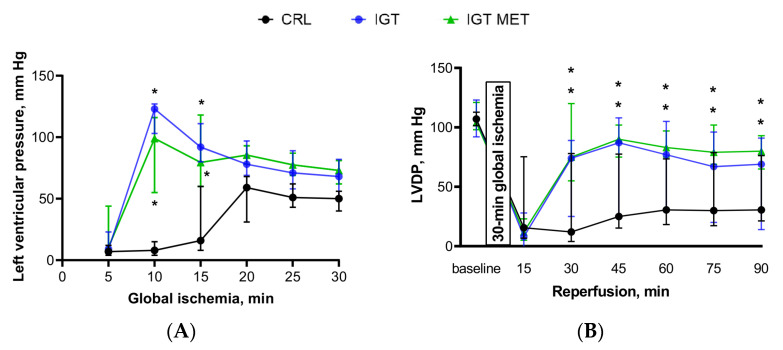
Functional parameters in isolated rat hearts were subjected to 30 min of global ischemia followed by 90 min of reperfusion. Notes: (**A**) Ischemic contracture (LV pressure —left ventricular pressure), (**B**) LVDP—left ventricular developed pressure, (**C**) LVEDP—left ventricular end-diastolic pressure, (**D**) CFR—coronary flow rate at baseline and during the experiment. Notes: Results are presented as median (25%; 75%). CRL—control group, IGT—impaired glucose tolerance group, IGT MET—impaired glucose tolerance + metformin group. * *p* < 0.05, both groups (IGT and IGT MET groups) in comparison with CRL both groups are separately marked (* < 0.05).

**Figure 3 ijms-23-06837-f003:**
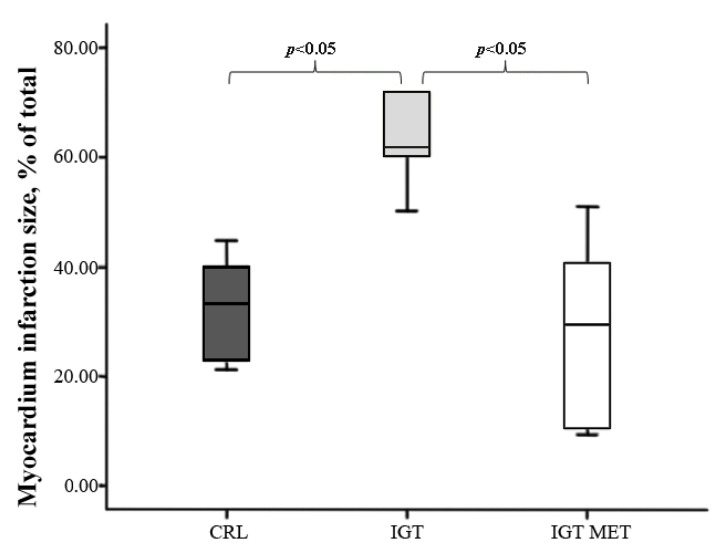
Myocardial infarction size in isolated rat hearts subjected to 30 min of global ischemia followed by 90 min of reperfusion. Notes: Results are presented as median (25%; 75%). CRL—control group, IGT—impaired glucose tolerance group, IGT MET—impaired glucose tolerance + metformin group.

**Figure 4 ijms-23-06837-f004:**
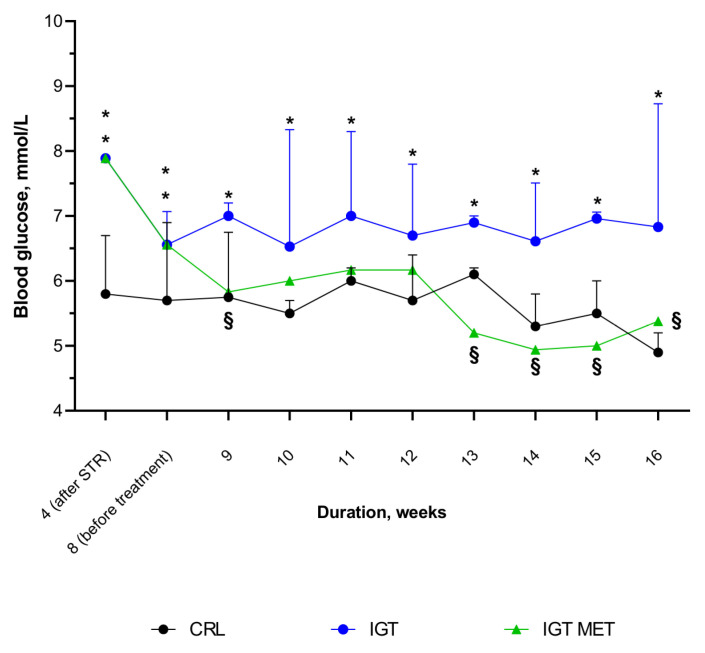
Dynamics of blood glucose level in rats from different groups. Notes: Results are presented as median (25%; 75%). C-CRL(control group), IGT—impaired glucose tolerance group, IGT MET—impaired glucose tolerance + metformin group. * *p* < 0.05, CRL both groups (IGT and IGT MET groups). in comparison with CRL group, § *p* < 0.05, in comparison with “IGT” group.

**Figure 5 ijms-23-06837-f005:**
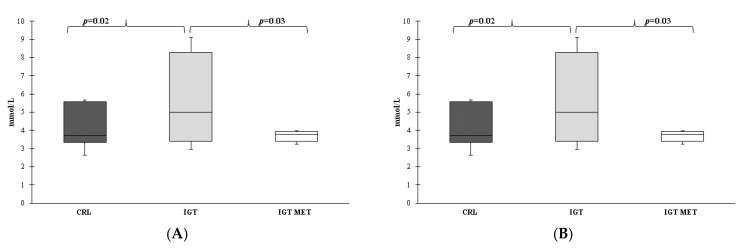
The lipid profile parameters of rats from different groups at the end of the experiment. Notes: (**A**) total cholesterol, (**B**) atherogenic coefficient. Results are presented as median (25%; 75%). CRL—control group, IGT—impaired glucose tolerance group, IGT MET—impaired glucose tolerance + metformin group. The atherogenic coefficient (AC) was calculated by the formula AC = (total cholesterol – HDL)/HDL.

**Figure 6 ijms-23-06837-f006:**
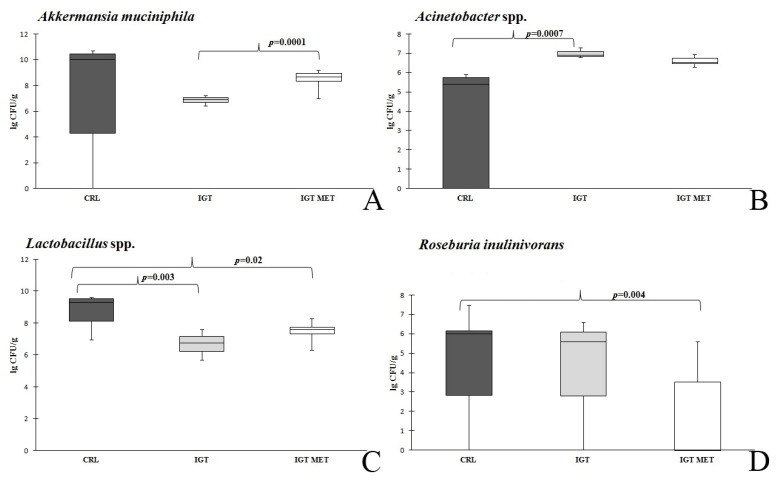
*Akkermansia muciniphila* (**A**), *Acinetobacter* spp. (**B**), *Lactobacillus* spp. (**C**) and *Roseburia unulinivorans* (**D**) quantitative content in the fecal samples of rats from different groups. Notes: Results are presented as median (25%; 75%). CRL—control group, IGT—impaired glucose tolerance group, IGT MET—impaired glucose tolerance + metformin group.

**Figure 7 ijms-23-06837-f007:**
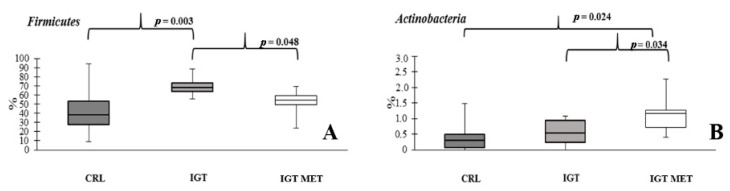
Relative abundances of phylums *Firmicutes* (**A**) and *Actinobacteria* (**B**) in the fecal samples of rats from different groups. CRL—control group, IGT—impaired glucose tolerance group, IGT MET—impaired glucose tolerance + metformin group.

**Figure 8 ijms-23-06837-f008:**
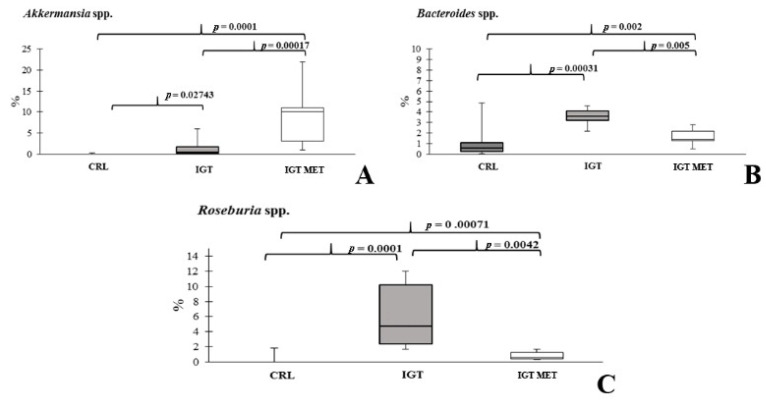
Relative abundances of genera *Akkermansia* (**A**), *Bacteroides* (**B**), and *Roseburia* (**C**) in the fecal samples of rats from different groups. CRL—control group, IGT—impaired glucose tolerance group, IGT MET—impaired glucose tolerance + metformin group.

**Figure 9 ijms-23-06837-f009:**
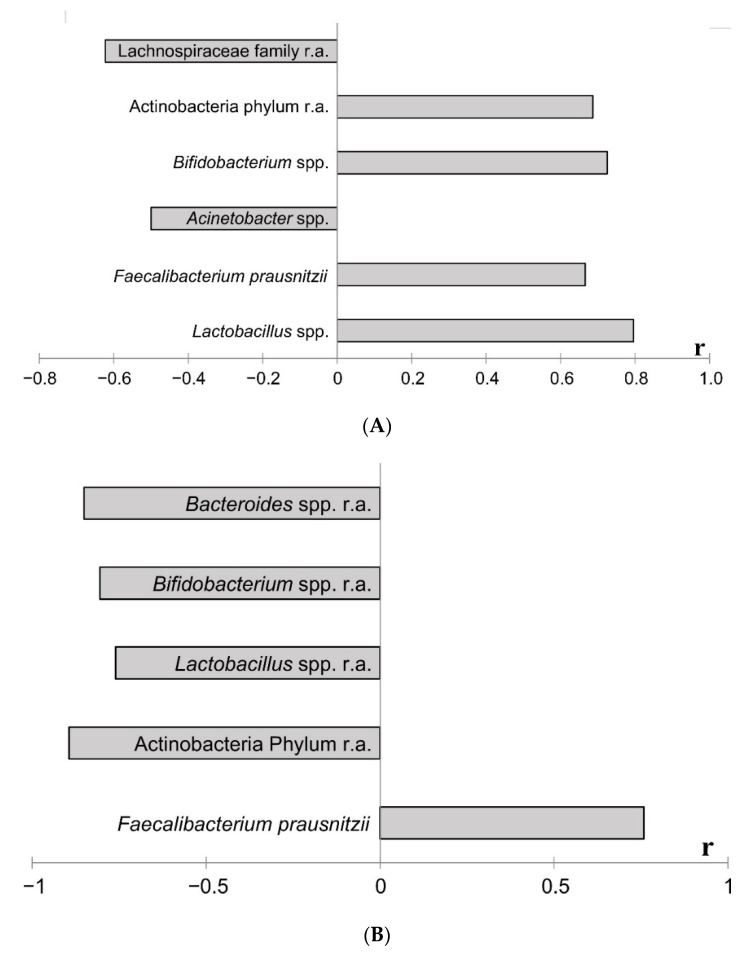
Coefficients of correlation by analysis between the relative abundance/quantitative content of *Akkermansia* spp. (**A**) or *Roseburia* spp. (**B**) and other taxa in the intestinal microbiota at the end of the experiment. Notes: total data for all animals, the results of the study are presented using *p* < 0.05. A search for correlations between the studied parameters was performed using Spearman’s test. r.a.—relative abundance. The quantitative content of taxa was analyzed in other causes.

**Figure 10 ijms-23-06837-f010:**
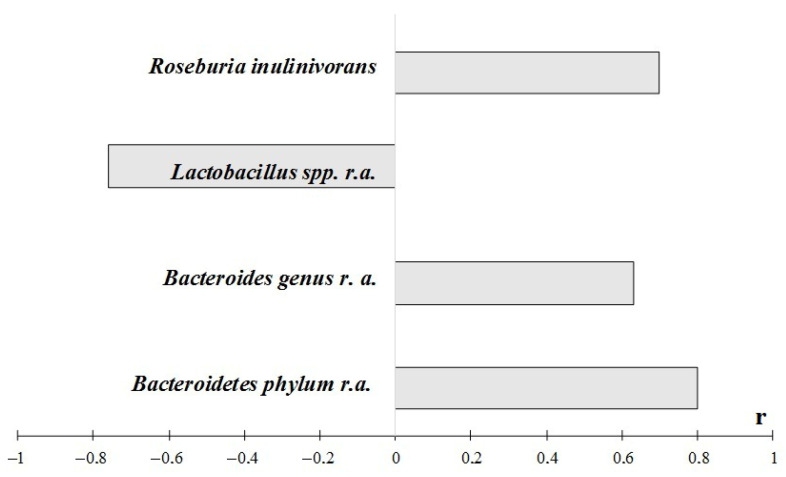
Coefficients of correlation between the quantity and relative abundance/quantitative content of bacterial taxa in intestinal microbiome and body weight at the end of the experiment. Notes: total data for all animals, the results of the study are presented using *p* < 0.05. A search for correlations between the studied parameters was performed using Spearman’s test. r.a.—relative abundance. The quantitative content of taxa was analyzed in other causes.

**Figure 11 ijms-23-06837-f011:**
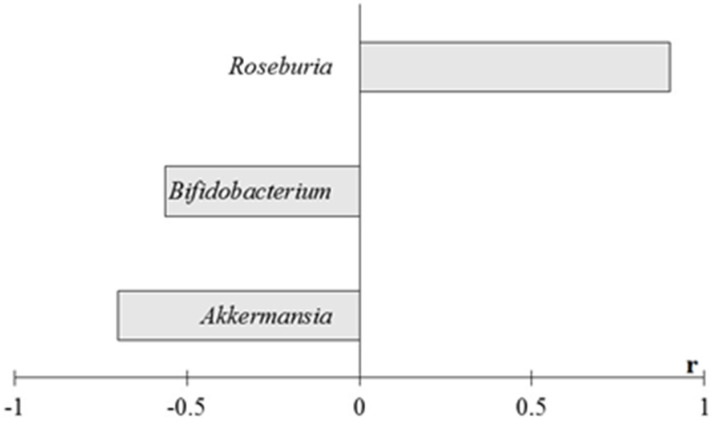
Coefficients of correlation between relative abundances of bacterial taxa in intestinal microbiome and BGL at the end of the experiment. Notes: total data for all animals, the results of the study are presented, using *p* < 0.05. A search for correlations between the studied parameters was performed using Spearman’s test.

**Figure 12 ijms-23-06837-f012:**
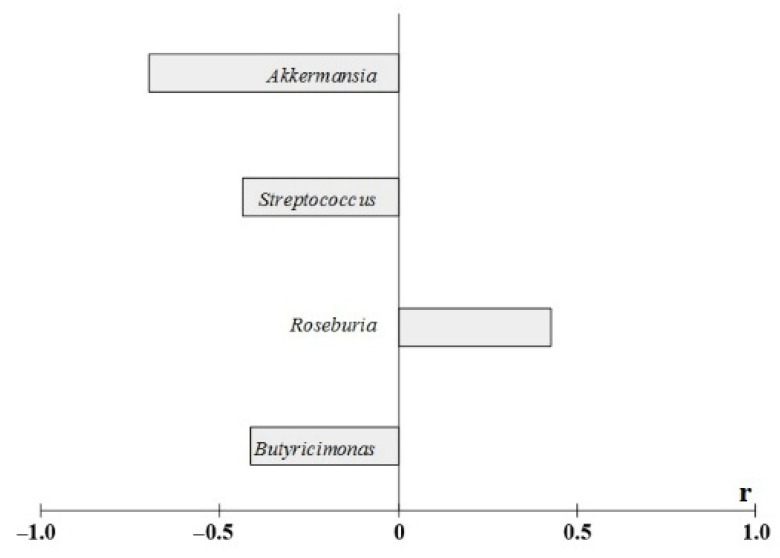
Coefficients of correlation between relative abundances bacterial taxa in the intestinal microbiome and myocardial infarction size at the end of the experiment. Notes: total data for all animals, the results of the study are presented using *p* < 0.05. A search for correlations between the studied parameters was performed using Spearman’s test.

**Table 1 ijms-23-06837-t001:** Study design. CRL—control group, IGT—impaired glucose tolerance group, IGT MET—impaired glucose tolerance + metformin group.

Groups	4 Weeks	Induction of IGT on 29th Day	4 Weeks	8 Weeks	Necropsy after 16 Weeks
**CRL**	Standard diet	-	Standard diet	Standard diet	Assessment of the composition of the intestinal microbiota. Isolated heart perfusion
**IGT**	High-fat diet	Injection of nicotinamide and streptozotocin	High-fat diet	High-fat diet	
**IGT MET**	High-fat diet	Injection of nicotinamide and streptozotocin	High-fat diet	High-fat diet + metformin treatment	

Body weight and food consumption measurement every 2nd day. BGL assessment on 28th, 56th, 63rd, 70th, 76th, 84th, 91st, 98th, 105th, and 112th days of the experiment at the same daytime (not fasting measurement). The blood samples were taken on the 112th day for additional biochemical study.

## Data Availability

Data available in a publicly accessible repository.

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
