# Peer review of "Metformin Influence on the Intestinal Microbiota and Organism of Rats with Metabolic Syndrome"

_ijms, 2022, doi:10.3390/ijms23126837_

Round 1
Reviewer 1 Report
In this experimental work, titled “Metformin Influence On The Intestinal Microbiota And Organism Of Rats With Metabolic Syndrome” Dr. Ermolenko and colleagues make use of a diffuse rat experimental model of IGT to verify the effect of metformin on several parameters metabolic syndrome and of CV function, including the myocardial infarction size after IR injury. The work aims at addressing a timely topic. The experimental system adopted is appropriate and data generally support the authors’ observations. Figures are self-explanatory and statistical analysis is generally appropriate. However, the work lacks originality, since it represents a validation of what is mostly already published (as fairly acknowledged by the authors). I frankly could not find any novelty, despite the robustness and the appropriateness of the observations presented. Further, as it is, this work does not add any mechanistic detail and remains merely of descriptive nature. Lastly, some of the conclusions of the authors (myocardial infarction size reduction on IGT met-treated rats) are challenged by previous studies in T2DM patients, appropriately sized (doi: 10.1161/JAHA.115.002314). This partially questions the translational relevance of this latter finding. In summary, this is a reasonably well-crafted work which however, only validates what has been already found without offering any further mechanistic insights. With regard to this latter point, the discussion does not offer any explanation or discussion of the results neither it does offer mechanistic insights even if of speculative nature.
Detail.
The text need revision, there are leftovers throughout the entire text: for example, at the end of the “statistical analysis” paragraph, there is a “RESULTS” word which should not be there.
Author Response
Dear Reviewer.
Thank you very much for interest, attention to our paper, a large and painstaking work related to its review. We are very grateful to you for your helpful comments and advice, which we tried to take into account. We will use them in our scientific work and when preparing manuscripts and reports.
Further, I attach answers to your questions and comments.
However, the work lacks originality, since it represents a validation of what is mostly already published (as fairly acknowledged by the authors). I frankly could not find any novelty, despite the robustness and the appropriateness of the observations presented.
It is studied that type 2 diabetes mellitus is accompanied by severe increase of cardio-vascular risk, leading to increased incidence of myocardial infarction and even cardiovascular death. On the other hand, data concerning IGT influence on cardiovascular morbidity and mortality are much less presented. Moreover, IGT impact on myocardial damage size is even less studied. Moreover, recently much attention has been paid to the hypothesis that pleotropic cardioprotective effects of metformin in diabetes mellitus could be connected with metformin influence on microbiota compound. On the other hand, similar effect in IGT conditions remains the subject of investigation, which, in our opinion, forms the novelty of our study. As to our knowledge, our study was the first to investigate correlation links between myocardial damage, carbohydrate and lipid metabolism disruption and microbiota compound in IGT conditions which helps to elucidate potential mechanisms of metformin cardioprotective effect.
Further, as it is, this work does not add any mechanistic detail and remains merely of descriptive nature.
Thank you, data concerning potential mechanism of metformin effects, interrelationships between microbiota changes and MS symptoms and myocardial damage have been included into the “Discussion” section.
Lastly, some of the conclusions of the authors (myocardial infarction size reduction on IGT met-treated rats) are by previous studies in T2DM patients, appropriately sized (doi: 10.1161/JAHA.115.002314).
The main difference of our work in comparison with others including doi: 10.1161/JAHA.115.002314 is that it focuses on IGT group, not a diabetic one. Data concerning cardioprotective effect of metformin in IGT conditions are lack.
There are studies focusing on metformin cardioprotective properties in patients with prediabetes and individuals without glucose metabolism impairment (doi: 10.1136/bmjdrc-2015-000090; doi: 10.2337/dc18-2356; doi: 10.1152/ajpheart.00054.2011), but these trials predominantly focus on the evaluation of surrogate cardiovascular risk markers. There are also some data about cardioprotective properties of metformin in animals without glucose metabolism impairments (doi: 10.1016/s1734-1140(12)70945-3, but we have not found experimental works evaluating infarct-limiting effect of metformin in prediabetic conditions. Therefore, we may assume that our scientific group was the first to study metformin cardioprotective potential in rat transient global myocardial ischemia. We previously investigated metformin influence on myocardial damage size and hemodynamic parameters in comparison with other glucose-lowering drug in IGT rats (doi: 10.1038/s41598-021-86132-2), while the aim of this study was to reveal possible mechanisms underlying metformin cardioprotective effect.
The text need revision, there are leftovers throughout the entire text: for example, at the end of the “statistical analysis” paragraph, there is a “RESULTS” word which should not be there.
We changed the location of the "Results" heading by moving it to the right place in the text.
In summary, this is a reasonably well-crafted work which however, only validates what has been already found without offering any further mechanistic insights. With regard to this latter point, the discussion does not offer any explanation or discussion of the results neither it does offer mechanistic insights even if of speculative nature.
We have added most of the mechanisms of metformin action and Akkermansia spp. to the discussion section,
The text need revision, there are leftovers throughout the entire text: for example, at the end of the “statistical analysis” paragraph, there is a “RESULTS” word which should not be there.
We have made a revision of the text. We corrected several typos and slightly changed the sentences stylistically. All corrections were marked in green. Also, notes explaining the reasons for corrections have been added to the text.
We have changed the location of the results category in accordance with the requirements for authors. We also made significant edits to the introduction sections, added Figures 5B and 9B. Most seriously changed the text of the discussion and added links. In addition, in accordance with your recommendation, the errors in citation were corrected.
Corrections are marked in green. Sincerely yours, Elena Ermolenko.

Reviewer 2 Report
The topic regarding Metformin Influence On The Intestinal Microbiota And Organism Of Rats With Metabolic Syndrome is not a new one. To provide a better manuscript please see my suggestions:
Please check the Instructions for authors regarding affiliation, type of the characters, draft for the manuscript, etc. The actual appearance of the manuscript looks sloppy, written in a hurry.
L62. Succession of references must be written as 13-19, not specifying each of them. Moreover, 2-3 references are enough to be cited for a statement, not 7.
L72. Aim of the study is not relevant, not presenting any new or special aspect regarding the topic. Please reshape and highlight it better.
Table 1. Please check the Instructions for authors regarding the shape of a Table.
Section 2. Too many very short subsections, with no relevance. A subsection must present and entire idea, well developed, not 1.5 lines. Please restructure from 2.6 to 2.9.
Not needing to divide 2.12.
Section 3. Results must be mentioned after L 207. Sloopy aspect.
Many results but the aspect part requires good improvement
All Figures 1-5 are blurred. Please provide better quality figures.
L365. Discussion section is 4 not 3.
L375-381 must be moved at the final of Discussion section, as the aim of the study. This aim of the study must be presented once!, at the final pf Introduction.
The Discussion chapter must be improved. Please discuss previous studies where A. muciniphila has been shown as an important bacterium for prevention of cardiovascular disease or metabolic syndrome, I suggest checking and referring to https://www.mdpi.com/2076-2607/9/3/618. Please make 2 graphical figures that resume the beneficial effects of Akkermansia muciniphila in metabolic syndrome and the role of metformin in metabolic syndrome (maybe you acan adapt Figure 3S to fit here). Please discuss the potential beneficial role of GLP-1 agonists in intestinal microbiota [https://www.spandidos-publications.com/10.3892/etm.2020.8714; https://doi.org/10.3390/diagnostics11061090 ]
After L479, please add the strengths and the weakness of your study.
List of Abbreviations must be removed. Respect the Instructions for authors and introduce all of them in the main manuscript: "Abbreviations must respect also the instructions for authors: Acronyms/Abbreviations/Initialisms have been defined the first time they appear in each of three sections: the abstract; the main text; under the first figure or table. When defined for the first time, the acronym/abbreviation/initialism should be added in parentheses after the written-out form". Please check and revise the entire manuscript in this regard.
Supplementary materials must be removed from the main manuscript and uploaded in the MDPI system in the dedicated section.
Author Response
Dear reviewer!
Thank you very much for interest, attention to our paper, a large and painstaking work related to its review. We are very grateful to you for your helpful comments and advice, which we tried to take into account. We will use them in our scientific work and when preparing manuscripts and reports.
Further, I attach answers to your questions and comments.
The topic regarding Metformin Influence On The Intestinal Microbiota And Organism Of Rats With Metabolic Syndrome is not a new one. To provide a better manuscript please see my suggestions: Please check the Instructions for authors regarding affiliation, type of the characters, draft for the manuscript, etc. The actual appearance of the manuscript looks sloppy, written in a hurry.
We checked the Instructions for authors again. After that, we changed the format of the table, improved the quality of the drawings, made changes to the heading of the text and corrected a number of section and unsuccessful sentences and phrases. The changes are highlighted with a green marker in the main text.
L62. Succession of references must be written as 13-19, not specifying each of them. Moreover, 2-3 references are enough to be cited for a statement, not 7.
We have made changes to the text and marked the links in accordance with the requirements of the publisher.
L72. Aim of the study is not relevant, not presenting any new or special aspect regarding the topic. Please reshape and highlight it better.
We have changed the wording of the aim in the introduction. «The aim of this study was to evaluate the influence of multifunctional antidiabetic drug metformin on body weight and metabolic parameters, including glycaemia, lipid profile, on the intestinal microbiota, as well as functional and morphological characteristics of myocardium under ischemia-reperfusion conditions in rats with IGT. We also performed a correlation analysis between the parameters that underwent changes after IGT induction and exposure to metformin to establish a causal relationship and search for new aspects of metformin action».
Table 1. Please check the Instructions for authors regarding the shape of a Table.
We have changed the table according to the requirements for the authors.
Section 2. Too many very short subsections, with no relevance. A subsection must present and entire idea, well developed, not 1.5 lines. Please restructure from 2.6 to 2.9. Not needing to divide 2.12.
We restructured subsection 2.6.-2.9. The 2.12.subsections names and numbers were deleted.
Section 3. Results must be mentioned after L 207. Sloopy aspect.
We changed the location of the "Results" heading by moving it to the right place in the text. All corrections are marked in green.
Many results but the aspect part requires good improvement
We changeв the aspect part due to significant changes in the discussion. Moreover, we added new date of statistical analysis and interpretation of some results. In particular, we have added a picture illustrating. the relationship between Acinetobacter (Figure 6S) and Roseburia (Figure 9B) and other taxa as well as additionally processed data on the lipid profile and revealed significant changes in the coefficient of atherogenicity of animals from different groups (Figure 5 B).
All Figures 1-5 are blurred. Please provide better quality figures.
We provided better quality figures 1-5.
L365. Discussion section is 4 not 3.
We have changed the number 3 to 4 in the numbering of the “Discussion” section.
L375-381 This aim of the study must be presented once!,at the final Introduction.
The aim of the study was moved at the final of “Discussion” section to the final part of the “Introduction” section. The old aim formulation has been removed.
The Discussion chapter must be improved.
Please discuss previous studies where A. muciniphila has been shown as an important bacterium for prevention of cardiovascular disease or metabolic syndrome,, I suggest checking and referring to https://www.mdpi.com/2076-2607/9/3/618.
We have added relevant comments to the final part of the discussion.
Please make 2 graphical figures that resume the beneficial effects of Akkermansia muciniphila in metabolic syndrome and the role of metformin in metabolic syndrome (may be you can adapt Figure 3S to fit here).
We made an additional figure reflecting the role of Akkermansia spp , whose population has increased, on individual components of the microbiota and clinical and laboratory manifestations of IGT. The figure has been added to the additional materials section (Figure 5S)
Please discuss the potential beneficial role of GLP-1 agonists in intestinal microbiota [https://www.spandidos-publications.com/10.3892/etm.2020.8714; https://doi.org/10.3390/diagnostics11061090 ]
We added information about influence of GLP-1 agonists and LPS in the section «Discussion».
After L479, please add the strengths and the weakness of your study.
The strong side of our study is the correlation analysis, which opens up new areas of research already more complete, taking into account the dynamics, the impact on the digestive and cardiovascular systems, immune and may be other (for axample nerves and endocrine systems). We focused on this in the final part of the discussion.
List of Abbreviations must be removed. Respect the Instructions for authors and introduce all of them in the main manuscript: "Abbreviations must respect also the instructions for authors: Acronyms/Abbreviations/Initialisms have been defined the first time they appear in each of three sections: the abstract; the main text; under the first figure or table. When defined for the first time, the acronym/abbreviation/initialism should be added in parentheses after the written-out form". Please check and revise the entire manuscript in this regard.
List of Abbreviations was deleted.
Supplementary materials must be removed from the main manuscript and uploaded in the MDPI system in the dedicated section.
Supplementary materials was removed from the main manuscript and was uploaded in the MDPI system in the dedicated section.
Sincerely, Elena Ermolenko.

Reviewer 3 Report
The authors carried out an interesting work to study the changes in the intestinal microbiota in IGT rats and the effect of long-term treatment with drug metformin on them. They also found new relationships between the intestinal microbiota in IGT rats and the severity of impairments in the isolated heart during perfusion and larger myocardium infarction, and also assessed the positive effect of metformin on resistance to myocardial dysfunction. The presented data are new and suitable statistics were used for their analysis. However, there are a number of comments and remarks, which are presented below.
Major comments
The authors present the results on the cardioprotective effect of metformin and discuss it in terms of changes in the microbiota and the pattern of pro-inflammatory factors. At the same time, they ignore the available data on other (and probably more significant) mechanisms of the cardioprotective effect of metformin. They should be presented and discussed. In addition, the data on inflammation factors levels are not given in the work, and it would be desirable to do so. It is also necessary to describe, as this is discussed in the article, how a change in the intestinal microorganisms studied in the work can affect inflammatory processes and biochemical parameters (if such data are available).
The authors focus on conflicting literature data on the problem they are studying of changes in the intestinal microflora in IGT and other metabolic disorders. But isn't this a consequence of different experimental conditions? Differences in food composition, bacterial insemination, use of antibiotics and other drugs that affect the survival of various strains of microorganisms can be crucial for the microbiota. What are these conditions in the other works analyzed by the authors?
Additionally, high dose metformin can lead to significant gastrointestinal dysfunctions, in which case many of the changes are secondary and will be highly dependent on the duration of therapy and the dose of metformin. Is it possible to draw common conclusions about the relationship between the microbiota and metformin therapy based on the use of a single dose of the drug and only one time period of the study? In this case, it would be useful to analyze the dynamics of microbiota changes starting from the first week. Moreover, weight loss and normalization of glucose levels were shown already in the first weeks of metformin treatment.
The description of the Section 2.4 is not entirely clear. The authors write that rats were first selected for the glucose range of 7.8-11.0 mmol/l, and then they do a glucose load test and again select rats with a glucose level in the same range. “We diagnosed IGT if BGL was in the interval of 7.8-11.0 at least at one time point and was not equal and did not exceed 11.1 mmol/L at any point”? At what point? What were the glucose levels in the first measurement (baseline?) and what were they in the second measurement (glucose stimulated?)? Explain.
What were the side effects of 8 weeks of treatment with a relatively high dose of metformin? Were there stomach upsets and lactic acidosis phenomena? Describe in more detail.
It is necessary to present the available literature data on how high doses of metformin affect the studied intestinal microorganisms (in the Discussion section).
Section 3.1.
The authors write that there is a difference in body weight at the start of metformin treatment, but there is no difference in Figure 1 at week 8. Thus, there is no obesity in rats of the IGT group.
Does not match description: “IGT induction led to significant body weight gain”. Write clearly.
Figure 4
The groups at the first points are mixed up. How the authors explain the suppression of glucose levels at the end of the experiment? Is this a result of metformin-induced lactic acidosis and gastrointestinal disorders? Describe the physical condition of the animals.
The lack of significant differences in triglyceride levels in the blood of control and IGT rats is a little surprising, especially with such a large difference in body weight and, presumably, with differences in adipose tissue mass. It is necessary to give data on triglycerides plasma levels, the ratio of LDL and HDL (atherogenic index) and the ratio of triglyceride/HDL cholesterol (predictors for insulin resistance), and also provide data on the adipose tissue mass, as an addition to Figure 1.
The authors discuss the development of insulin resistance in IGT rats, but the present study does not provide sufficient evidence for this, based only on glucose levels. It is not clear when the glucose was measured, and what level of insulin was in the rats.
The same must be provided for the metformin IGT group.
The paragraph “We aimed to evaluate the influence...” refers to the objectives of the study and is not appropriate in the Discussion section.
Minor remarks
Line 42 AMPK is an energy sensor, but not “the liver sensor”
Lines 42-43 5’-adenosine monophosphate-activated protein kinase (AMPK), but not “adenosine monophosphate protein kinase”
Author Response

(The authors gave the same response as above.)

Round 2
Reviewer 1 Report
The author have, overall, improved significantly the manuscript by promptly reacting to the raised issues.
Reviewer 2 Report
The article is very well written and respects all the requirements of an original article. i recommend publication.